# Psychological functioning in survivors of COVID-19: Evidence from recognition of fearful facial expressions

Federica Scarpina[1,2]*, Marco Godi[3], Stefano Corna[3], Ionathan Seitanidis[4], Paolo Capodaglio[4,5], Alessandro Mauro[1,2]

**1** Istituto Auxologico Italiano, IRCCS, U.O. di Neurologia e Neuroriabilitazione, Ospedale S. Giuseppe, Piancavallo, Italy, **2** "Rita Levi Montalcini" Department of Neurosciences, University of Turin, Turin, Italy, **3** Institute of Veruno, Istituti Clinici Scientifici Maugeri IRCCS, Gattico-Veruno, Italy, **4** Istituto Auxologico Italiano, IRCCS, U.O. di U.O. Riabilitazione Osteoarticolare, Ospedale S. Giuseppe, Piancavallo, Italy, **5** Department of Surgical Sciences, Physical and Rehabilitation Medicine, University of Turin, Turin, Italy

* federica.scarpina@unito.it, f.scarpina@auxologico.it

**Data Availability Statement:** All data files are available from the database (http://doi.org/10.5281/zenodo.4898818).

## Abstract

Evidence about the psychological functioning in individuals who survived the COVID-19 infectious is still rare in the literature. In this paper, we investigated fearful facial expressions recognition, as a behavioural means to assess psychological functioning. From May 15th, 2020 to January 30th, 2021, we enrolled sixty Italian individuals admitted in multiple Italian COVID-19 post-intensive care units. The detection and recognition of fearful facial expressions were assessed through an experimental task grounded on an attentional mechanism (i.e., the redundant target effect). According to the results, our participants showed an altered behaviour in detecting and recognizing fearful expressions. Specifically, their performance was in disagreement with the expected behavioural effect. Our study suggested altered processing of fearful expressions in individuals who survived the COVID-19 infectious. Such a difficulty might represent a crucial sign of psychological distress and it should be addressed in tailored psychological interventions in rehabilitative settings and after discharge.

## Introduction

During the Coronavirus disease 2019 (COVID-19) worldwide pandemic, individuals report psychological distress (i.e., [1–5]). However, evidence about the psychological functioning in those who survived the COVID-19 infectious is rare, as well as most of the research is still ongoing.

During the pandemic, hospitalized patients are isolated, experiencing substantial reduced social interactions for extended periods. Severe restrictions are applied in the intensive and post-intensive care units: patients are required to stay in their rooms; social interaction among patients, groups, family and caregivers, and group therapies, are not allowed. When possible, the number of personnel is minimized. Finally, surgical masks are worn by the patients and

**Funding:** The authors received no specific funding for this work.

**Competing interests:** The authors have declared that no competing interests exist.

the therapists. Sun and colleagues [6] describe the individuals' psychological experience of COVID-19 during hospitalization: patients report feelings of loneliness and self-isolation; moreover, in the early stages of the disease, they report negative emotional attitudes towards it, which include fear and denial, as well as stigma [6]. This qualitative description seems to agree with the first preliminary evidence of a higher level of anxiety and somatization symptoms [7] and a higher level of post-traumatic stress symptoms and depressive symptoms [1, 4] in these patients. Research about the effect of COVID-19 on mental health is still ongoing. However, it is highly needed to improve treatment and mental health care planning [4].

In this work, we assessed the fear-related psychological functioning in a sample of individuals who were hospitalized in Italian COVID-19 post-intensive care units. We used a behavioural approach, applying the short version [8] of the implicit facial emotion recognition task [9, 10] grounded on the attentional mechanism of the redundant target effect [11, 12]. Considering the extensive literature about this attentional effect [i.e., 11, 12], the approach proposed in this paper is convenient to provide a strong *a-priori* hypothesis. We focused on the emotion of fear [13, 14]. This primary emotion is strictly linked to the anxiety symptoms phenomenology [15, 16]. Moreover, fear and anxiety share the cerebral circuits (i.e., [14]). Altered recognition of facial emotion expressions has been suggested to be a sign of mental health difficulties [17–19] and psychological distress [14].

## Materials and methods

This study was approved by the Ethical Commission of the IRCCS Istituto Auxologico Italiano. It was performed accordingly to the Declaration of Helsinki's principles [20]. The entire study was scripted through the free software OpenSesame [21]. The experiment was run on a laptop. The duration of the procedure was of around five minutes.

### Participants

In this cross-sectional study, participants had been recruited at the COVID-19 post-intensive care units of the involved Institutions from from May 15th, 2020 to January 30th, 2021. These inclusion criteria have been applied: 1) adult participants (aged > 18 years); 2) previously hospitalizion in the COVID-19 intensive care unites; 3) clinical stability defined by ability to perform bedside active mobilization without a reduction of oxygen saturation (SpO2) below 90%; 4) complete weaning from sedative and antipsychotic drugs. These exclusion criteria had been applied: 1) respiratory distress signs; 2) need of respiratory support with a fraction of inspired oxygen (FiO2) >60%; 3) need of continuous positive airway pressure devices; 4) signs of cardiovascular instability; 5) positive anamnesi for neurological and psychiatric diagnosis; 7) signs of dementia.

Overall, sixty participants were enrolled. 58.3% of participants was male. About age, 6.7% of participants was in the range 31–45 years; 18.3%, in the range 46–60 years; and 75%, in the range 61 years and over. These results seemed to be in line with Sheehy [22] about older ages in individuals hospitalized because of COVID-19 rehabilitation. The majority of the participants were right-handed (76.7%), while a small percentange of individuals reported to be left-handed (11.7%) and ambidextrous (11.6%). About the level of education, 31.7% participants reported five years of schooling; 53.3%, eight years; 23.3%, twelve years; 15%, sixteen years or more.

Participants answered a short survey according to a four-point Likert scale questionnaire designed to explore the subjective perception of the psychological functioning and the level of empathy, at the time of the experiment [8]. The list of questions is reported in Table 1. The majority of the participants described a lower state of tension or upset, and a moderate level of

**Table 1. Psychological description.**

| In this moment: | 1 not at all | 2 not much | 3 somewhat | 4 very much | Statistical results |
|---|---|---|---|---|---|
| I feel calm | 5.1% | 13.6% | 66.1% | 15.3% | $\chi^2 = 54.55; \boldsymbol{p} < \mathbf{0.001}$ |
| I feel tense | 91.7% | 5% | 1.7% | 1.7% | $\chi^2 = 142.4; \boldsymbol{p} < \mathbf{0.001}$ |
| I feel upset | 67.8% | 20.3% | 8.5% | 3.4% | $\chi^2 = 91.08; \boldsymbol{p} < \mathbf{0.001}$ |
| I feel relaxed | 15% | 33.3% | 43.3% | 8.3% | $\chi^2 = 18.8; \boldsymbol{p} < \mathbf{0.001}$ |
| I feel happy | 22.4% | 19% | 37.9% | 20.7% | $\chi^2 = 5.3; p = 0.15$ |
| I feel worried | 28.3% | 43.3% | 20% | 8.3% | $\chi^2 = 15.6; \boldsymbol{p} < \mathbf{0.001}$ |
| I feel emphatic | 15% | 18.3% | 50% | 16.7% | $\chi^2 = 20.13; \boldsymbol{p} < \mathbf{0.001}$ |
| I feel feelings that I cannot identify | 45% | 25% | 21.7% | 8.3% | $\chi^2 = 16.53; \boldsymbol{p} < \mathbf{0.001}$ |
| People around me appear more anxious/afraid than usually | 31.7% | 30% | 28.3% | 10% | $\chi^2 = 7.33; p < 0.06$ |

For each of the four-step questions relative to the psychological functioning, we report the percentage (%) of respondents. We used the chi-square ($\chi$2) test to determine a statistically significant difference between the observed frequencies in the steps for each psychological question. In bold, significant p-value ($\leq$ 0.05). N = 60.

worries. Most of them reported being moderately calm and relax; also, they perceived them-selves as moderately empathic. A higher percentage of participants reported to be adequately conscious of their feelings; however, some participants expressed some concerns about this ability.

## Experimental task

We used the short version [8] of the implicit facial emotion recognition task [8–10] focused on the emotion of fear. It was a go–no go task, grounded on the redundant target effect [11]: individuals respond faster when two identical targets are presented simultaneously rather than when presented alone; moreover, the competitive presence of a distractor (that is another emotion or a neutral expression) affects the correct recognition of the target.

Photographs of a male face and a female face with a fearful expression [23] were shown in four different experimental conditions: *(i) single*: the fearful face was presented on the right OR left of a fixation cross; *(ii) congruent*: the fearful face was presented simultaneously on the right AND left of the fixation cross; *(iii) emotional incongruent*: the fearful face was presented on the right OR left of the fixation cross along with a different negative emotion (i.e., anger), or *(iv) neutral incongruent*: the target was presented on the right OR left of the fixation cross along with a neutral expression. Overall, the task consisted of 40 trials. For each experimental condition, eight trials were shown; thus, 32 valid trials were tested. Moreover, eight catch trials were randomly presented. Specifically, in two catch trials, we showed the neutral expression; in the other two, the angry expression; in the other four trials, we showed the neutral expression contrasted with the angry expression (two trials), and two angry expressions presented simultaneously on the right and left of the fixation cross (two trials).

In each trial, pictures were shown for 350 milliseconds; participants had a maximum of 1500 milliseconds from the onset of the visual stimuli to provide an answer. The inter-stimulus interval varied randomly between 650 and 950 milliseconds. Participants were invited to respond as soon as they noticed a fearful expression, pressing a key (i.e., the letter h) on the keyboard.

Reaction time in millisecond and accuracy were collected for valid trials.

## Analyses

We removed from the entire experiment participants who reported more than four *false alarms* (i.e., they answered in when no target was shown, in the catch trials), or an overall level

of accuracy in the valid trials less than 10%, which might indicate that the participant was randomly guessing his/her responses. Also, we removed from the analyses answers provided over the threshold of 1000 milliseconds (which might indicate lack of attention) and below the threshold of 50 milliseconds (i.e., impulsive responses) were not considered in the analyses [8, 9].

*Reaction Time* in milliseconds from the stimulus (i.e., when the target, meaning the emotion of fear, was correctly detected) and the *percentage of accuracy* (i.e., the percentage of correct answers to the valid trials) were computed for each of the four experimental conditions. We analyzed RT and percentage of accuracy analyzed. We run a repeated-measure ANOVA with the within-subjects factor of *Condition* (four levels: single, congruent, emotional incongruent, neutral incongruent) to probe the main hypothesis of this study. Estimated marginal mean comparisons Bonferroni-corrected were applied in the case of a significant main effect. We performed a priori analysis through the software G*Power [24]: a total sample size of 36 participants would be required to achieve a power (1—β) of 0.95, setting the α value at 0.05.

We also calculated and analyzed the efficiency score [25], which combines accuracy and latencies as the average of correct RT divided by the proportion of correct responses, for each experimental conditions. Indeed, attentional difficulties as well as mental fatigue are described in people after COVID-19 [26, 27]. Moreover, we asked our participants to respond as soon as possible, with a possible effect on the level of accuracy. Thus, IES score were was analyzed through a repeated-measure ANOVA with the within-subjects factor of *Condition* (four levels: single, congruent, emotional incongruent, neutral incongruent).

Since the pandemic and its global consequences on individuals' psychological functioning, we cannot collected *ad-hoc* control sample for this study. Thus, we compared the present sample's performance with previous data available in the literature [9], as done in Scarpina [8]. Specifically, for both the reaction time and the level of accuracy, an independent sample t-test was performed independently for each experimental condition between the performance registered in this experiment and the performance reported in Scarpina and colleagues [9], in which twenty-five healthy subjects (16 women, *Age* M = 42 years; SD = 14; range 23–61; *Education* M = 15 years; SD = 2; range: 8–18) were tested with a long extended version of the task.

## Results

### Preprocessing data

Fifteen participants were excluded from the analyses because of the data preprocessing: six of them reported more than four *false alarms* (i.e., they answered in the case of a catch trial, meaning when no target was shown). Nine participants were excluded since their overall level of accuracy for the valid trials was less than 10%. Moreover, 2.98% of valid trials were excluded because answers were provided over the threshold of 1000 milliseconds and below the threshold of 50 milliseconds. Preliminary inspection of the raw data showed the presence of an outlier. However, we decided not to remove it, in line with the previous studies [8–10].

### Detection of fearful expressions: Reaction time

The main effect of *Condition* was significant [$F(3, 90) = 13.04$; $p < 0.001$; partial $\eta^2 = 0.3$]. The post hoc comparisons showed a significant lower reaction time in the neutral incongruent condition in comparison with all the other conditions [$p \leq 0.001$]. The other comparisons were not significant [$p \geq 0.42$] (Fig 1).

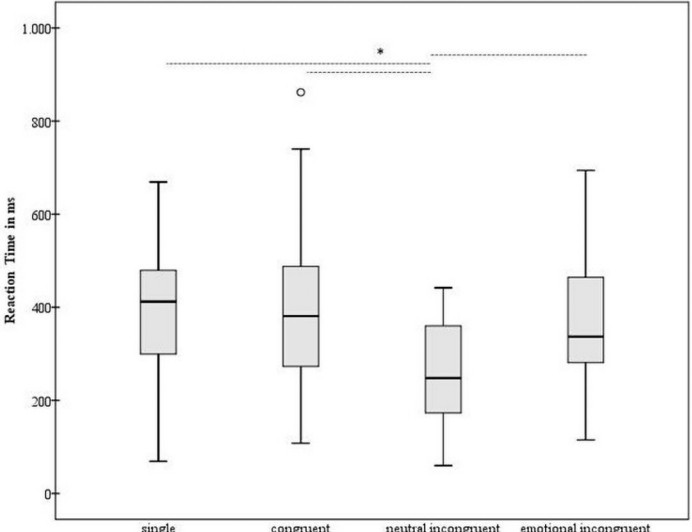

**Fig 1. Detection of fearful expressions: Reaction time.** The mean of reaction time expressed in millisecond (y-axis—ms) is reported for each experimental condition (x-axis: single, congruent, neutral incongruent, emotional incongruent). We show the minimum, the lower quartile, the median, the upper quartile, the maximum, and the outliers are shown. Dotted horizontal lines indicate the significant difference between conditions, according to the main analyses; * p value ≤ 0.001.

## Recognition of fearful expressions: Level of accuracy

The main effect of *Condition* was significant [$F(3, 132) = 4.27$; $p = 0.006$; partial $\eta^2 = 0.08$]. The post hoc comparisons showed a significant level of accuracy in the neutral incongruent condition in comparison with the congruent condition [$p = 0.01$]. The other comparisons were not significant [$p \geq 0.14$] (Fig 2).

## The efficiency score

When we analyzed our data considering the trade-off between accuracy and velocity, i.e. the efficiency score, no significant main effect of *Condition* (single M = 8.71; SD = 6.44; congruent M = 10.21; SD = 12.2; neutral incongruent M = 6.48; SD = 4; emotional incongruent M = 8.38; SD = 5.8) [$F(3,90) = 1.72$; $p = 0.16$; partial $\eta^2 = 0.05$] emerged. Thus, the previous results represented a trade-off between accuracy and velocity. Crucially, the absence of any difference between conditions confirmed the absence of the redundant target effect in our participant's performance.

## Comparison with previous data

As shown in Fig 3 and reported in Table 2, our sample of individuals in post-acute COVID-19 show a different performance when compared with the sample of healthy individuals described in Scarpina and colleagues [9]. Specifically, the present sample was faster in detecting fearful expression when showed together with neutral expressions (i.e., neutral incongruent condition): this behavior was not in line with the redundant target effect, according to which people should be slower in incongruent conditions in comparison with the congruent and single conditions. Moreover, our participants reported a significantly lower level of accuracy in recognizing fearful expressions in all the experimental conditions.

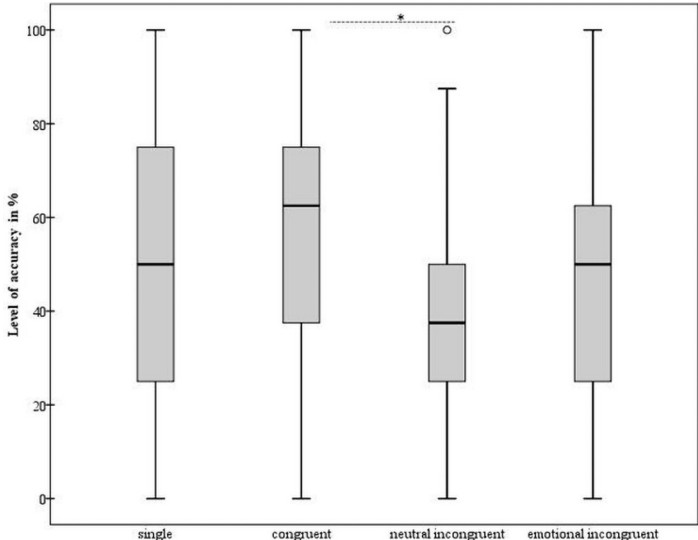

**Fig 2. Recognition of fearful expressions: Level of accuracy.** The level of accuracy expressed in percentage (y-axis–%) is reported for each experimental condition (x-axis: single, congruent, neutral incongruent, emotional incongruent). We show the minimum, the lower quartile, the median, the upper quartile, the maximum, and the outliers. Dotted horizontal lines indicate the significant difference between conditions, according to the main analyses; * p value $\leq$ 0.05.

## Discussion

In this work, we investigated the processing of fearful facial expressions in individuals recovered in COVID-19 post-intensive care units, through a behavioral approach. As reported in previous studies (i.e., [9, 10, 28]), an altered recognition of facial emotion expressions represents a sign of mental health difficulties [18, 19] and psychological distress [14]. We observed an alteration of this process in our sample: specifically, our participants reported difficulties in detecting and recognizing fearful expressions.

The task used in this experiment grounded on the attentional phenomenon known as redundant target effect [11, 12], according to which individuals are more efficient in detecting (i.e., the reaction time) and recognizing (i.e., the level of accuracy) emotional expressions when two identical faces, rather than only one face, are shown. Moreover, the presence of a face showing a contrasting emotion or a neutral expression impacts the performance. Our participants reported a performance that was not in line with this effect [11, 12]. Focusing on the level of accuracy we might observe that in our participant there was no behavioural advantage in recognizing fearful expressions when two identical faces (congruent condition), rather than only one face (single condition), were shown. Moreover, no advantage emerged when the easier (single and congruent) conditions were contrasted with the more difficult (neutral and emotional) incongruent ones. The level of accuracy was comparable across the experimental conditions, except for the neutral one, in which participants reported a lower percentage of response. Focusing on the reaction time, our participants were faster in detecting fearful expression in the neutral congruent condition, although no difference emerged between the other experimental conditions. Thus, the presence of a contrasting neutral expression seemed to increase the target detection and recognition. The ability to ignore irrelevant information (in this case, the neutral stimulus) is directly related to the load in the processing of the relevant information (i.e., the fearful expression) [29]. However, it should be considered that a lack of attentional resources, as well as mental fatigue, has been reported as neuropsychological

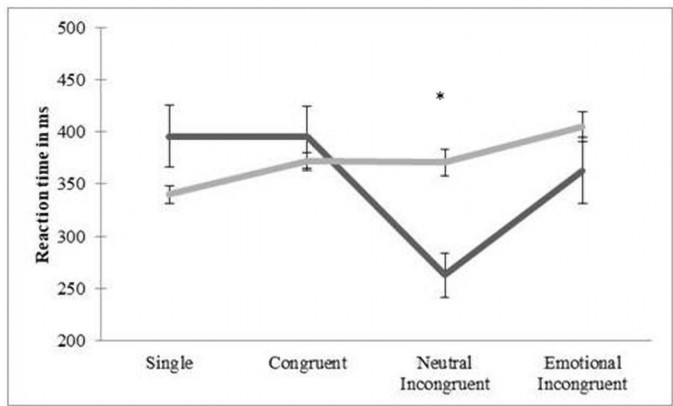

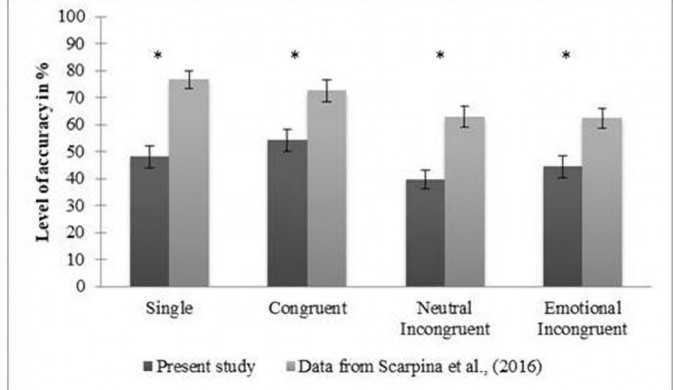

**Fig 3. Comparison with previous data. Upper part.** Mean (lines) and standard error (vertical lines) relative to reaction time expressed in milliseconds (y-axis—ms) for each experimental condition (x-axis: single, congruent, emotional incongruent, neutral incongruent) is reported for the sample (n = 45) of the present experiment (dark gray lines) and the sample (n = 20) in Scarpina and colleagues (2018) (light gray lines). **Below part.** Mean (bars) and the standard error (vertical lines) relative to the level of accuracy expressed in percentage (y-axis–%) for each experimental condition (x-axis: single, congruent, emotional incongruent, neutral incongruent) are reported for the sample (n = 45) of the present experiment (dark gray bars) and the sample (n = 25) in Scarpina and colleagues (2018) (light gray bars). * p value < 0.05.

sequelae associated with COVID-19 [26–27], affecting cognitive performance. When we calculated and analyzed the efficiency score [25], the results showed a trade-off between accuracy and velocity. Crucially, the results also confirmed the absence of the redundant target effect in our participants' performance, suggesting the difficulties in processing correctly fearful expressions. A decreased performance in our sample emerged also when we compared it with the previous data relative to healthy individuals collected before the pandemic [9]. However, some cautions should be taken in the interpretation of the comparison between our sample and the controls from Scarpina and colleagues [9], since our participants were older. Older adults have increased difficulty in recognizing some of the basic emotions, specifically anger and sadness, which were not tested in our experiment [30]. On the other hand, the redundant target effect emerges in the elderly: specifically, older individuals generally show a greater advantage of bilateral presentations relative to unilateral presentations, and thus an increased redundant target effect, in comparison with youngers [31]. Since this evidence, the absence of the redundant target effect in our participants' performance as an effect of ageing seemed an unlikely explanation.

**Table 2. Comparison with previous data.**

| | Single | Congruent | Neutral Incongruent | Emotional Incongruent |
|---|---|---|---|---|
| | **Reaction time in millisecond** | | | |
| **Present study** | 396 (197) | 395 (198) | 263 (143) | 363 (213) |
| **Scarpina et al., (2018) [9]** | 340 (42) | 372 (42) | 371 (64) | 405 (73) |
| | t = 1.83; p = 0.007; d' = 0.39; 95% CI[-4.92;116.95] | t = 0.74; p = 0.45; d' = 0.16; 95%CI [-38.23;84.23] | t = 3.57; p< **0.001**; d' = 0.97; 95% CI[-157.61;58.38] | t = 0.9; p = 0.34; d' = 0.26; 95%CI [-11.73;27.73] |
| | **Level of accuracy in percentage** | | | |
| **Present study** | 48.06 (27.3) | 54.17 (26.65) | 39.72 (23.28) | 44.44 (26.19) |
| **Scarpina et al., (2018) [9]** | 76.62 (16.8) | 72.5 (20.68) | 63 (19.83) | 62.31 (18.93) |
| | t = 4.76; **p<0.001**; d' = 1.26; 95%CI [-39.09M-18.02] | t = 2.97; **p = 0.004**; d' = 0.76; 95% CI[-29.77;-6.88] | t = 4.21; **p<0.001**; d' = 1.07; 95%CI [-33.79;-12.76] | t = 2.98; **p = 0.003**; d' = 0.78; 95% CI[-28.72;-7.01] |

Mean and standard deviation (in brackets) relative to reaction time (expressed in milliseconds in the upper part) and the level of accuracy (expressed in percentage in the lower part) are reported for each experimental condition (single, congruent, neutral incongruent, emotional incongruent), for the sample of participants of this study (N = 45) and the sample of participants reported in Scarpina and colleagues (2018) (N = 25). We used an independent sample t-test (degrees of freedom = 68) to verify any possible difference between the two samples. We report t-value, p-value, Cohen's d', and the 95% standard symmetric confidence interval (CI). In bold, significant result (p value ≤ 0.05).

In this study, we tested only the emotion of fear. Indeed, the severe restrictions applied in the COVID-19 post-intensive care units, as well as the possible higher level of fatigue experienced by patients, limited us in applying a longer experimental task to test all the basic emotions [13]. Thus, it cannot be established if the difficulty in processing emotional facial expressions registered in our experiment pertained only to fear, suggesting an emotion-specific difficulty, or if it might spread to other emotions, suggesting an overall emotional impairment. We also underline that in this paper we cannot establish the origin of this alteration, such as the experience of confinement and social restriction during the hospitalization, or possible effects of the disease on the brain and cognition. However, when the same experimental task was performed by a sample of Italian individuals during the first lockdown (April 12nd, 2020 to May 3rd, 2020) who declared no COVID-19 related symptoms, no alteration in the psychological behaviour—at least at the level of accuracy—was reported [8].

## Conclusions

The recognition of altered emotional processing in patients who survived COVID-19 might be crucial to detect precociously signs and symptoms of psychological distress [28, 29]. These patients might need even more psychological support than typical rehabilitative care units patients because of the possible higher levels of survivors' guilt and post-traumatic stress disorder [32]. Considering the role played by facial emotional recognition in social interaction and psychological well-being, the effect of prolonged hospitalization in the case of COVID-19 might be further explored and eventually targeted in psychosocial interventions in post-intensive care and after discharge.

## Author Contributions

**Conceptualization:** Federica Scarpina.

**Data curation:** Federica Scarpina.

**Formal analysis:** Federica Scarpina.

**Investigation:** Federica Scarpina, Marco Godi, Stefano Corna, Ionathan Seitanidis, Paolo
Capodaglio.

**Methodology:** Federica Scarpina.

**Supervision:** Paolo Capodaglio, Alessandro Mauro.

**Writing – original draft:** Federica Scarpina.

**Writing – review & editing:** Paolo Capodaglio, Alessandro Mauro.

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
