## [Decision Letter · Decision Letter 0]

20 May 2021

PONE-D-21-04797

Psychological functioning in survivors of COVID-19: evidence from ability to recognize expressions of fear.

PLOS ONE

Dear Dr. Scarpina,

Thank you for submitting your manuscript to PLOS ONE. After careful consideration, we feel that it has merit but does not fully meet PLOS ONE’s publication criteria as it currently stands. Therefore, we invite you to submit a revised version of the manuscript that addresses the points raised during the review process.

We look forward to receiving your revised manuscript.

Kind regards,

Zezhi Li, Ph.D., M.D.

Academic Editor

PLOS ONE

Journal Requirements:

2. Please note that PLOS ONE does not copy edit accepted manuscripts (https://journals.plos.org/plosone/s/criteria-for-publication#loc-5). To that effect, please ensure that your submission is free of typos and grammatical errors.

4.We note that you have indicated that data from this study are available upon request. PLOS only allows data to be available upon request if there are legal or ethical restrictions on sharing data publicly. For more information on unacceptable data access restrictions, please see http://journals.plos.org/plosone/s/data-availability#loc-unacceptable-data-access-restrictions.

Reviewers' comments:

Reviewer's Responses to Questions

**Comments to the Author**

1. Is the manuscript technically sound, and do the data support the conclusions?

Reviewer #1: Yes

Reviewer #2: Yes

2. Has the statistical analysis been performed appropriately and rigorously? 

Reviewer #1: Yes

Reviewer #2: Yes

3. Have the authors made all data underlying the findings in their manuscript fully available?

Reviewer #1: No

Reviewer #2: Yes

4. Is the manuscript presented in an intelligible fashion and written in standard English?

Reviewer #1: Yes

Reviewer #2: Yes

5. Review Comments to the Author

Reviewer #1: The objective of the research is to assess the processing of fear-related facial expressions, as a behavioral means to assess psychological functioning in individuals who survived to the COVID-19. The data showed an altered behavior in detecting and recognizing fearful expressions. Specifically, their performance was in disagreement with the expected behavioral performance. Although this study is quite interesting and these results provide evidence about the psychological functioning in individuals who survived to the COVID-19 infectious, the following issues need to be addressed before it is accepted and published.

Points to address:

• When comparing the reaction time and accuracy of the present sample’s performance with previous Scarpina’s data in the literature, as the age and education of the two samples are very different, 75% of the current study’s subjects are over 61 which is reasonable because of COVID-19 rehabilitation. I wonder why not pick the age and education matched subjects to make a fair comparison? I am curious to see whether that make a difference or not.

• Need more explanation and discussion on why the reaction time in the neutral incongruent condition is significantly lower in comparison with all the other conditions.

• In page 7 Experimental task, valid trials in which answers provided over the threshold of 1000 milliseconds and below the threshold of 50 milliseconds were not considered in the analyses. What’s the rationale to exclude trials over 1000 ms, is there a reference on this threshold? As 75% of the subjects are over 61, I wonder whether 1000 ms is too strict for older people. What percent of trails that got reject due to RT > 1000 ms?

• In page 13 last paragraph, I don’t understand what this sentence means “…according which in the case of incongruent stimulation, people should be lower in comparison with the congruent and single conditions.” People should be lower? You mean slower?

• Table 1 looks not that great, please make it tighter.

• Is there a specific reason to keep the outlier subject in the analysis for Figure 1 and 2. Why not just remove this outlier?

Reviewer #2: I think your paper should undergo another revision.

Hereunder I'll try to summarize my main concerns.

The most important are those regarding methodology.

Regarding the "language" group, please keep in mind that I'm italian and I suffer from the main two errors/bias I think we have regarding English, which lead to the following (questionable) rules of thumb: (1) "if it sounds similar to a literal translation from Italian, then it is wrong", (2) use subordinate or parentetical clauses, long sentences, passive or impersonal passive only if it is really necessary; (3) there's a tradeoff between precision and redundancy.

If you do not agree, you can skip this “language concerns group”.

Nevertheless I suggest that you proofread the paper again; if it is possible with the help of a native speaker.

Should your paper not be immediately published, next time, please, add row numbers to help referencing to the text.

I avoid using locutions as "in my opinion" etc.: I apologize for any perceived lack of politeness.

Good luck

[1] Methodology and statistics

[1.1] Data about “subjective psychological functioning”. (1) you do not use them: you do not compare the “healthy” group on it, you do not use any kind of “dependence measure” with othe variables in the covid group; (2) you do not balance for order effect (but you do not use them, so it is not a big deal).

[1.2] Stimuli. Stimuli should differ only for the content of the variables values they contain/represent. You have four levels of your main independent variable:

only one content [fear + nothing]

twice the same content [fear + fear]

one content and one distracto [fear + neutral]

one content and one strong distractor [fear + anger]

the “nothing” part of the first should have the same visual characteristics of the others, without the emotional-face content. Imagine your stimuli face images are 200x200 gray levels [0, 255] bitmap. You can sum the total level of gray (the sum of the 2*10^4 numbers in the range 0, 255) and build random 200x200 control image with the same total “grayness”:

[1.3] Signal to noise ratio.

We can represent your stimuli in the following way:

[pictures]:

subject: 1, 2, 3, 4, … N

gender: {F, M}

emotion: {neutral, anger, fear}

so each picture can be represented with the 3-uple: 〈s, g=""〉 where S is the id of the subject, G her or his gender, E her or his emotion.

Each stimulus can be described as

element_A: a picture 〈s, g=""〉

element_B a picture 〈s, g=""〉 or nothing (only the background)

order: element_A on the right and element_B on the left; or viceversa.

You should give us at least the following information:

1- how many different subjects are represented in the pictures?

2- how did you build “catch stimuli”? neutral+nothing, neutral+neutral, neutral+anger and anger+anger?

3- you have a priori 80% noise+signal (32/40) and 20% (8/40) only noise. There’s a “response strategy” for a subject who does not understand the task and is slow enough to respond (press the button say 60% of times) in the time window [end_of_stimulus+50, end_of_simulus_+950] such that he would have a high accuracy without being discarded? In other words which is the baseline of a random response? Give us some information of how to understand an unbalanced response schema as the one you use. What is a good score? This is important because you can have bad scores which are statistically different one from the other - but they are still bad scores.

[2] Language

To use and moderate the following concerns, please remind what I said at the very beginning of this review. Hereunder I’ll write a few examples (these are not the only ones: please proofread the paper again) of what I think is wrong on the language side.

“[...] we investigated the capability to recognize fearful facial expressions […]"

vs

“[…] we investigated fearful facial expressions recognition […]" (or whatever without "capability of")

“redundant target effect [11-12], according which individuals”

vs

“redundant target effect [11-12], according to which individuals” (but for this we need a native speaker)

“people should be lower in comparison”

vs

“people should be slower//obtain lower scores//whatever…”

“in bold, in the case of significant p-value (≤ 0.05)” (you do not need "in the case of": check other articles)

“About the level of accuracy, our participants report a significant lower performance in all the experimental conditions, suggesting a lower efficacy in recognizing fearful expressions.” What does this mean? The “suggesting” part is useless. Similarly in “In this work, we focused on the automatic, unintentional and unconscious mechanism of processing facial expressions of fear [13-14].” Are there unconscious mechanisms which are both not automatic and intentional? Is there something automatic which is intentional? Can something unintenional be anything else but not-automatic?

“At the light of this consideration” (this it the prototype of what I meant).

“the difference was significantly different only”

“might suggest” you do not need to temper twice

“origin of such an alteration” vs “origin of this alteration” or “of this kind of alteration” etc.

“Considering the role playing by ” ("role playing" vs "role played by" but please try to find another locution)

“social restriction because the hospitalization” vs “[…] because of [...]”

“patients who survived to COVID-19 might be”: I think “survive” is not with “to”; please check it out.〈/s,〉〈/s,〉〈/s,〉

6. PLOS authors have the option to publish the peer review history of their article (what does this mean?). If published, this will include your full peer review and any attached files.

Reviewer #1: No

Reviewer #2: No

---

## [Author Response · Author response to Decision Letter 0]

4 Jun 2021

---- Answers to Reviewers ---

Reviewer #1

The objective of the research is to assess the processing of fear-related facial expressions, as a behavioral means to assess psychological functioning in individuals who survived to the COVID-19. The data showed an altered behavior in detecting and recognizing fearful expressions. Specifically, their performance was in disagreement with the expected behavioral performance. Although this study is quite interesting and these results provide evidence about the psychological functioning in individuals who survived to the COVID-19 infectious, the following issues need to be addressed before it is accepted and published.

REPLY: We thank the Reviewer for the time devoted to reviewing our article. We answered to all the questions and we modified the main text accordingly to them. Changes are highlighted in yellow in the main text. 

Points to address:

• When comparing the reaction time and accuracy of the present sample’s performance with previous Scarpina’s data in the literature, as the age and education of the two samples are very different, 75% of the current study’s subjects are over 61 which is reasonable because of COVID-19 rehabilitation. I wonder why not pick the age and education matched subjects to make a fair comparison? I am curious to see whether that make a difference or not.

REPLY: We agree with the Reviewer that our sample was older than the controls. Unfortunately, we were not allowed to enroll a control group for this experiment because of the pandemic; thus, we used published data. 

In this comment, the Reviewer raised an interesting question about the role of ageing in facial emotion recognition. One possibility to answer the Reviewer’s question would be to perform an ANCOVA with Age (and eventually Education) as covariates. However, this solution cannot be pursued: indeed, how we have collected the demographical data (through categorical questions) in this experiment was different from what reported in Scarpina and colleagues (2018) (i.e., quantitative factors). 

However, previous evidence reported in the literature assisted us to provide an answer to this question. The redundant target effect is generally observed in the elderly: Linnet and Rosen (2012) reported that not only older participants show the effect in the same direction of youngers, but crucially they show a greater advantage of bilateral presentation relative to unilateral presentation; in other words, older participants show an increased redundant target effect, in comparison with youngers (Linnet and Roser, 2012). Instead, our participants did not show the effect. Thus, the absence of a redundant target effect because of ageing in our sample seemed an unlikely explanation. 

We were not able to find any previous evidence relative to the role of schooling on the redundant target effect. However, in Scarpina (2019), in which the task was used for testing remotely healthy participants during the first lockdown, it was observed that schooling interacted with the level of accuracy in recognizing fearful expression only for those individuals with 18 years of attended schooling or more. In our sample, only 15% of participants reported sixteen years or more of schooling. 

The topic about ageing was very interesting since it clarified our results better. For this reason, we included a new paragraph about this topic in our Discussion.

Lines 280-289. A decreased performance in our sample emerged also when we compared it with the previous data relative to healthy individuals collected before the pandemic [9]. However, some cautions should be taken in the interpretation of the comparisons between our sample and the controls from Scarpina and colleagues [9]: indeed, our participants were older. Older adults have increased difficulty in recognizing some of the basic emotions, specifically anger and sadness, which were not tested in our experiment [29]. On the other hand, the redundant target effect emerges in the elderly: specifically, older individuals generally show a greater advantage of bilateral presentations relative to unilateral presentations, and thus an increased redundant target effect, in comparison with youngers [30]. Since this evidence, the absence of the redundant target effect in our participants’ performance as an effect of ageing seemed an unlikely explanation. 

• Need more explanation and discussion on why the reaction time in the neutral incongruent condition is significantly lower in comparison with all the other conditions.

REPLY: This was a very interesting comment that helped us in clarifying our data. As we discussed in our revised version of our manuscript, the presence of a contrasting neutral expression increased the detection of the target. It is known that the ability to ignore irrelevant information (the neutral stimulus) is directly related to the load in the processing of the relevant information (the fearful expression) (Lavie, 1995); nevertheless, a lack of attentional resources, as well as mental fatigue, has been recognized as a sign of the neuropsychological sequelae associated to COVID-19, with a possible effect on the performance in our task. To verify such an effect, we took advantage of the efficiency score (Townsend and Ashby, 1983) which combines accuracy and latencies as the average of correct RT divided by the proportion of correct responses (lines 150-160). As we reported at lines 208-213, we might observe the presence of a trade-off between accuracy and velocity. Crucially, no difference between conditions emerged, confirming the absence of the redundant target effect in our participant’s performance. 

• In page 7 Experimental task, valid trials in which answers provided over the threshold of 1000 milliseconds and below the threshold of 50 milliseconds were not considered in the analyses. What’s the rationale to exclude trials over 1000 ms, is there a reference on this threshold? As 75% of the subjects are over 61, I wonder whether 1000 ms is too strict for older people. What percent of trails that got reject due to RT > 1000 ms?

REPLY: Thank you for this comment. In the new version of our paper, we reported that we removed from the analyses answers provided over the threshold of 1000 milliseconds, because they might suggest lack of attention, and below the threshold of 50 milliseconds, because they might suggest impulsive responses. The thresholds were applied in line with the seminal article [8-9]. 

Lines 139-144. We removed from the entire experiment participants who reported more than four false alarms (i.e., they answered in when no target was shown, in the catch trials), or an overall level of accuracy in the valid trials less than 10%, which might suggest that the participant was randomly guessing his/her responses. Also, we removed from the analyses answers provided over the threshold of 1000 milliseconds (which might suggest lack of attention) and below the threshold of 50 milliseconds (i.e., impulsive responses) were not considered in the analyses [8-9].

In the section relative to the preprocessing data, we reported how many participants/ how many trials were excluded. 

Lines 172-178. Fifteen participants were excluded from the analyses because of the data preprocessing: six of them reported more than four false alarms (i.e., they answered in the case of a catch trial, meaning when no target was shown). Nine participants were excluded since their overall level of accuracy for the valid trials was less than 10%. Moreover, 2.98% of valid trials were excluded because answers were provided over the threshold of 1000 milliseconds and below the threshold of 50 milliseconds. Preliminary inspection of the raw data showed the presence of an outlier. However, we decided not to remove it, in line with the previous studies [8-10].

Thus, the percentage of trials excluded because of the threshold of 1000 ms and 50 ms was very low (2.98%) and in line with the previous literature in the field. 

 • In page 13 last paragraph, I don’t understand what this sentence means “…according which in the case of incongruent stimulation, people should be lower in comparison with the congruent and single conditions.” People should be lower? You mean slower?

REPLY: We thank the Reviewer for having noticed this inconsistency. In the new version of our paper, we modified the sentence as follows:

Lines 218-223. Specifically, the present sample was faster in detecting fearful expression when showed together with neutral expressions (i.e., neutral incongruent condition): this behavior was not in line with the redundant target effect, according to which people should be slower in incongruent conditions in comparison with the congruent and single conditions. Moreover, our participants reported a significantly lower level of accuracy in recognizing fearful expressions in all the experimental conditions.

• Table 1 looks not that great, please make it tighter.

REPLY: We thank the Reviewer for this comment. In the new version of our paper, Table 1 was modified. 

• Is there a specific reason to keep the outlier subject in the analysis for Figure 1 and 2. Why not just remove this outlier?

REPLY: A preliminary inspection of the raw data showed the presence of this outlier. However, we decided not to remove it, as done in Scarpina (2020) and more specifically in Scarpina et al. (2018), that was the study from which we extracted the data relative to the controls. We clarified this point in our revised version of the paper. 

Lines 177-178. Preliminary inspection of the raw data showed the presence of an outlier. However, we decided not to remove it, in line with the previous studies [8-10].

Reviewer #2. 

I think your paper should undergo another revision.

Hereunder I'll try to summarize my main concerns.

REPLY: REPLY: We thank the Reviewer for the time devoted to reviewing our article. We answered all the questions and we modified the main text accordingly to them. Changes are highlighted in yellow in the main text. 

The most important are those regarding methodology.

Regarding the "language" group, please keep in mind that I'm italian and I suffer from the main two errors/bias I think we have regarding English, which lead to the following (questionable) rules of thumb: (1) "if it sounds similar to a literal translation from Italian, then it is wrong", (2) use subordinate or parentetical clauses, long sentences, passive or impersonal passive only if it is really necessary; (3) there's a tradeoff between precision and redundancy.

If you do not agree, you can skip this “language concerns group”.

Nevertheless I suggest that you proofread the paper again; if it is possible with the help of a native speaker.

REPLY: We thank the Reviewer for these valuable suggestions. We profoundly revised the writing of our article. We are sorry for the typos in the first version of our paper. 

Should your paper not be immediately published, next time, please, add row numbers to help referencing to the text.

REPLY: We apologize for any inconvenience because of this lack. In the new version of our manuscript, we used the row numbers. 

I avoid using locutions as "in my opinion" etc.: I apologize for any perceived lack of politeness.

Good luck

[1] Methodology and statistics

[1.1] Data about “subjective psychological functioning”. (1) you do not use them: you do not compare the “healthy” group on it, you do not use any kind of “dependence measure” with othe variables in the covid group; (2) you do not balance for order effect (but you do not use them, so it is not a big deal).

REPLY: We thank the Reviewer for this comment. We used the psychological questions to sketch abruptly the psychological functioning of our participants at the time of the experiment (since no other structured information from psychological questionnaires was available). Since we did not use this information in our main analyses, we moved this part in the section relative to the “Participants”, as suggested by the Reviewer. 

[1.2] Stimuli. Stimuli should differ only for the content of the variables values they contain/represent. You have four levels of your main independent variable:

only one content [fear + nothing]

twice the same content [fear + fear]

one content and one distracto [fear + neutral]

one content and one strong distractor [fear + anger]

the “nothing” part of the first should have the same visual characteristics of the others, without the emotional-face content. Imagine your stimuli face images are 200x200 gray levels [0, 255] bitmap. You can sum the total level of gray (the sum of the 2*10^4 numbers in the range 0, 255) and build random 200x200 control image with the same total “grayness”:

REPLY: The question arisen by the Reviewer was very interesting. This question mainly focused on the visual characteristics of stimuli. This aspect should be strictly controlled in the case of an experiment in which the visual factor is crucial. However, this is not our case, since we manipulated an attentional (and not a visual) component. The redundant target effect refers to the advantage in the performance that is obtained with multiple redundant-stimulus presentations when compared with the presentations of a single stimulus.

It should be noticed that the methodology applied in this study is not new. Indeed, it grounds on a very extensive previous literature in cognitive psychology, about which we offer a short overview. Our research group have published three articles in which the same experiment was used: 

● Scarpina, F., Varallo, G., Castelnuovo, G., Capodaglio, P., Molinari, E., & Mauro, A. (2021). Implicit facial emotion recognition of fear and anger in obesity. Eating and Weight Disorders-Studies on Anorexia, Bulimia and Obesity, 26(4), 1243-1251.

● Scarpina, F. (2020). Detection and Recognition of Fearful Facial Expressions During the Coronavirus Disease (COVID-19) Pandemic in an Italian Sample: An Online Experiment. Frontiers in Psychology, 11.

● Scarpina, F., Melzi, L., Castelnuovo, G., Mauro, A., Marzoli, S. B., & Molinari, E. (2018). Explicit and implicit components of the emotional processing in non-organic vision loss: behavioral evidence about the role of fear in functional blindness. Frontiers in psychology, 9, 494.

Moreover, the following articles provided from other groups of research should be consulted: they represented the seminal works about the redundant target effect.

● Miniussi, C., Girelli, M., & Marzi, C. A. (1998). Neural site of the redundant target effect: Electrophysiological evidence. Journal of cognitive neuroscience, 10(2), 216-230.

● Gondan, M., Niederhaus, B., Rösler, F., & Röder, B. (2005). Multisensory processing in the redundant-target effect: a behavioral and event-related potential study. Perception & psychophysics, 67(4), 713-726.

● Tamietto, M., Corazzini, L. L., de Gelder, B., & Geminiani, G. (2006). Functional asymmetry and interhemispheric cooperation in the perception of emotions from facial expressions. Experimental brain research, 171(3), 389-404.

● Tamietto, M., & de Gelder, B. (2008). Affective blindsight in the intact brain: Neural interhemispheric summation for unseen fearful expressions. Neuropsychologia, 46(3), 820-828.

[1.3] Signal to noise ratio.

We can represent your stimuli in the following way:

[pictures]:

subject: 1, 2, 3, 4, … N

gender: {F, M}

emotion: {neutral, anger, fear}

so each picture can be represented with the 3-uple: where S is the id of the subject, G her or his gender, E her or his emotion.

Each stimulus can be described as

element_A: a picture

element_B a picture or nothing (only the background)

order: element_A on the right and element_B on the left; or viceversa.

REPLY: We are sorry, but we are not able to entirely understand this answer/suggestion. However, we would attempt to answer it. However, we will be ready to answer it with more details if we missed the point. 

Perhaps, the Reviewer has been suggesting to consider the emotion of anger. In our previous works (Scarpina et al., 2018, 2021), we explored both the emotion of fear and anger through this method. However, it is a go-no go task, in which participants are asked to focus on one emotion at a time. So, to test two emotions (such as fear and anger), we have to enlarge the number of trials and assess separately the two emotions in different blocks (as done in our previous works). However, it was not convenient in this study; indeed, considering the contextual limitations expressed in the paper, a priori we decided to focus on the emotion of fear because of its meaning about psychological functioning. 

If the Reviewer has been suggesting to consider the role of gender’s faces or the role of spatial position on the target detection and recognition, we did not consider these two effects in our main analyses because of the previous evidence. In all the previous studies performed by our groups, we did not register these components affecting the redundant target effect performance, in line with other studies (i.e., please refer to the list of papers reported previously). Introducing these factors in our analyses would shift the Rearders’ attention from the main point, that is the redundant target effect. Moreover, the statistical analyses used in this paper was the same applied in the previous studies relative to the same experiment to maximize the data comparability.

You should give us at least the following information:

In the following part, we furnished all the information required by the Reviewer about our experiment. 

1- how many different subjects are represented in the pictures?

REPLY: At the line 121 of the revised version of our manuscript, we specified that the photographs of a male face and a female face were used, as done in our previous works. Also, we detailed the origin of these photographs [ref. 13].

2- how did you build “catch stimuli”? neutral+nothing, neutral+neutral, neutral+anger and anger+anger?

REPLY: We specified how we built the catch trials in the main manuscript, as required by the Reviewer.

Lines 128-132. Moreover, eight catch trials were randomly presented Specifically, in two catch trials, we showed the neutral expression; in the other two, the angry expression; in the other four trials, we showed the neutral expression contrasted with the angry expression (two trials), and two angry expressions presented simultaneously on the right and left of the fixation cross (two trials). 

3- you have a priori 80% noise+signal (32/40) and 20% (8/40) only noise. There’s a “response strategy” for a subject who does not understand the task and is slow enough to respond (press the button say 60% of times) in the time window [end_of_stimulus+50, end_of_simulus_+950] such that he would have a high accuracy without being discarded? In other words which is the baseline of a random response? Give us some information of how to understand an unbalanced response schema as the one you use. What is a good score? This is important because you can have bad scores which are statistically different one from the other - but they are still bad scores.

REPLY: The schema used to construct the experiment presented in this work is the same proposed in the previous articles in the fields (please, refer to the previous list). Specifically, it should be noticed that in the redundant target effect, the performance was rated across the four experimental conditions (in which, someone might consider the single stimulation or the congruent stimulation as baseline). The catch trials were used to check if the participant had correctly understood the experiment’s request (i.e., push the button every time he/she has seen the target). Perhaps, if a participant adopts the strategy to push a button in each trial, he/she reported a higher number of responses also in the catch trials, as well as in the valid trials. For this reason, as we specified in the main text, we removed from the entire experiment participants who reported more than four false alarms (i.e., they answered in when no target was shown, in the catch trials). As reported in the results, six of participants reported more than four false alarms, and they were excluded from the analyses. 

[2] Language

To use and moderate the following concerns, please remind what I said at the very beginning of this review. Hereunder I’ll write a few examples (these are not the only ones: please proofread the paper again) of what I think is wrong on the language side.

REPLY: We thank the Reviewer for the attention devoted to reviewing our manuscript. We revised our paper to avoid typos. Also, all the following changes were included. 

“[...] we investigated the capability to recognize fearful facial expressions […]"

vs

“[…] we investigated fearful facial expressions recognition […]" (or whatever without "capability of")

REPLY: Done. Thank you. 

“redundant target effect [11-12], according which individuals”

vs

“redundant target effect [11-12], according to which individuals” (but for this we need a native speaker)

REPLY: Done. Thank you.

“people should be lower in comparison”

vs

“people should be slower//obtain lower scores//whatever…”

REPLY: Done. Thank you.

“in bold, in the case of significant p-value (≤ 0.05)” (you do not need "in the case of": check other articles)

REPLY: Done. Thank you.

“About the level of accuracy, our participants report a significant lower performance in all the experimental conditions, suggesting a lower efficacy in recognizing fearful expressions.” What does this mean? The “suggesting” part is useless. 

REPLY: We thank the Reviewer for this suggestion. 

Similarly in “In this work, we focused on the automatic, unintentional and unconscious mechanism of processing facial expressions of fear [13-14].” Are there unconscious mechanisms which are both not automatic and intentional? Is there something automatic which is intentional? Can something unintenional be anything else but not-automatic?

REPLY: We thank the Reviewer to have pointed out this possible theoretical criticism. Of course, it is out of the scope of this manuscript to detail the role of consciousness and intentionality in decision-making about emotional stimuli. To avoid any confusion, we did not include any reference to these concepts in our paper. 

“At the light of this consideration” (this it the prototype of what I meant).

REPLY: Done. Thank you.

“the difference was significantly different only”

REPLY: Done. Thank you.

“might suggest” you do not need to temper twice

REPLY: Done. Thank you.

“origin of such an alteration” vs “origin of this alteration” or “of this kind of alteration” etc.

REPLY: Done. Thank you. 

“Considering the role playing by ” ("role playing" vs "role played by" but please try to find another locution)

REPLY: Done. Thank you.

“social restriction because the hospitalization” vs “[…] because of [...]”

REPLY: We changed it. Thank you

“patients who survived to COVID-19 might be”: I think “survive” is not with “to”; please check it out.

---

## [Decision Letter · Decision Letter 1]

28 Jun 2021

Psychological functioning in survivors of COVID-19: evidence from recognition of fearful facial expressions.

PONE-D-21-04797R1

Dear Dr. Scarpina,

We’re pleased to inform you that your manuscript has been judged scientifically suitable for publication and will be formally accepted for publication once it meets all outstanding technical requirements.

Kind regards,

Zezhi Li, Ph.D., M.D.

Academic Editor

PLOS ONE

Additional Editor Comments (optional):

Reviewers' comments:

Reviewer's Responses to Questions

**Comments to the Author**

1. If the authors have adequately addressed your comments raised in a previous round of review and you feel that this manuscript is now acceptable for publication, you may indicate that here to bypass the “Comments to the Author” section, enter your conflict of interest statement in the “Confidential to Editor” section, and submit your "Accept" recommendation.

Reviewer #1: All comments have been addressed

Reviewer #2: All comments have been addressed

2. Is the manuscript technically sound, and do the data support the conclusions?

Reviewer #1: Yes

Reviewer #2: Yes

3. Has the statistical analysis been performed appropriately and rigorously? 

Reviewer #1: Yes

Reviewer #2: Yes

4. Have the authors made all data underlying the findings in their manuscript fully available?

Reviewer #1: No

Reviewer #2: Yes

5. Is the manuscript presented in an intelligible fashion and written in standard English?

Reviewer #1: Yes

Reviewer #2: Yes

6. Review Comments to the Author

Reviewer #1: The authors have satisfactorily responded to all my questions and made the necessary changes to the manuscript. The revised version of the manuscript appears to be good. I do not have any further questions.

Reviewer #2: About the method I'm still not convinced about stimuli construction and signal to noise ratio.

Line 111: probably you need among instead of between

7. PLOS authors have the option to publish the peer review history of their article (what does this mean?). If published, this will include your full peer review and any attached files.

Reviewer #1: No

Reviewer #2: No

---

## [Editor Report · Acceptance letter]

9 Jul 2021

PONE-D-21-04797R1 

Psychological functioning in survivors of COVID-19: evidence from recognition of fearful facial expressions. 

Dear Dr. Scarpina:

I'm pleased to inform you that your manuscript has been deemed suitable for publication in PLOS ONE. Congratulations! Your manuscript is now with our production department. 

Kind regards, 

on behalf of

Dr. Zezhi Li 

Academic Editor

PLOS ONE